# Vision-Based Deep Reinforcement Learning of UAV-UGV Collaborative Landing Policy Using Automatic Curriculum

Chang Wang [1], Jiaqing Wang [2,*], Changyun Wei [2], Yi Zhu [3], Dong Yin [1] and Jie Li [1]

1 College of Intelligence Science and Technology, National University of Defense Technology, Changsha 410073, China; wangchang07@nudt.edu.cn (C.W.); yindong@nudt.edu.cn (D.Y.); lijie09@nudt.edu.cn (J.L.)
2 College of Mechanical and Electrical Engineering, Hohai University, Changzhou 213022, China; c.wei@hhu.edu.cn
3 School of Computer Science, Nanjing Audit University, Nanjing 211800, China; 270352@nau.edu.cn
* Correspondence: j.wang@hhu.edu.cn

**Abstract:** Collaborative autonomous landing of a quadrotor Unmanned Aerial Vehicle (UAV) on a moving Unmanned Ground Vehicle (UGV) presents challenges due to the need for accurate real-time tracking of the UGV and the adjustment for the landing policy. To address this challenge, we propose a progressive learning framework for generating an optimal landing policy based on vision without the need of communication between the UAV and the UGV. First, we propose the Landing Vision System (LVS) to offer rapid localization and pose estimation of the UGV. Then, we design an Automatic Curriculum Learning (ACL) approach to learn the landing tasks under different conditions of UGV motions and wind interference. Specifically, we introduce a neural network-based difficulty discriminator to schedule the landing tasks according to their levels of difficulty. Our method achieves a higher landing success rate and accuracy compared with the state-of-the-art TD3 reinforcement learning algorithm.

**Keywords:** deep reinforcement learning; automatic curriculum learning; UAV landing





## 1. Introduction

Recent years have witnessed the significant potential of air–ground collaboration systems due to the improvement of levels of autonomy in robotics. The use of air and ground unmanned vehicles has become an important goal for real-world tasks of search-and-rescue scenarios [1–3]. Specifically, Unmanned Aerial Vehicles (UAVs) could significantly assist Unmanned Ground Vehicles (UGVs) by providing localization data and serving as communication relays [4]. However, a UAV has limited payload capacity and endurance. To address these limitations, UGVs are often utilized as mobile platforms for UAV recharging and maintenance. Therefore, UAV autonomous landing techniques have become essential for effective UAV-UGV collaboration.

Many approaches have been proposed to address the autonomous landing problem such as Fuzzy control [5], Model Predictive Control (MPC) [6], PD (Proportional, Derivative) control [7] , PID (Proportional, Integral, Derivative) control [8]. However, UAV autonomous landing remains a challenge because the landing process can be easily interrupted by unexpected UGV acceleration, direction change, or wind interference [9]. As one solution, vision-based control methods can be used to track and locate the UGV, resulting in the use of vision combined with reinforcement learning to address the landing task [10–13]. In our previous work [11], we integrated a PID controller with the Deep Deterministic Policy Gradient (DDPG) [14] algorithm. By introducing an agent to automatically tune parameters during the landing process, we achieved successful landings on a static marker. However, landing in dynamic environments with interference remains a challenge, and it is typically time-consuming to train the landing policy with a difficult level of task setting.

The Automatic Curriculum Learning (ACL) paradigm, as discussed in [15], breaks down a complex task into a series of simpler subtasks, initiating the training process with manageable conditions that progressively intensify in difficulty. There have been numerous efforts to integrate ACL with Deep Reinforcement Learning (DRL). Ren et al. [16] proposed a self-paced curriculum learning method, filtering transitions based on coverage penalty. Several ACL-based methods for robot control have been proven effective. For instance, the challenge of UAV mapless navigation was segmented into three task stages, with the task difficulty dynamically escalating according to the agent's navigation proficiency [17]. In a similar work, Hu et al. [18] suggested employing handcrafted Curriculum Learning (CL) to enhance experience sampling for fixed-wing aircraft motion control. Moreover, the evolutionary Nav-Q curriculum learning framework seamlessly integrated predicted Q-value insights with ACL to tackle DRL problems [19]. These studies highlight that implementing curriculum learning across the agent population not only significantly reduces convergence time but also enhances policy performance. However, we note that poorly designed curricula can lead to inefficient learning, disrupting the initial stages of DRL training by imposing excessively challenging tasks [20].

To address the aforementioned challenge, we introduce the Land-Automatic Curriculum Learning (Land-ACL) method. Automatic Curriculum Learning (ACL) has garnered significant attention because of its capacity to provide comprehensive signals for policy training. Recent studies conducted in [21–23] further substantiate the effectiveness of ACL in both simulated robot experiments and field studies. We design Land-ACL to facilitate the autonomous landing of a UAV on a moving platform using the Twin Delayed Deterministic Policy Gradient (TD3) algorithm, as outlined in [24]. We use the TD3 algorithm as the motion controller for the UAV as it is renowned for its effectiveness in robotic control problems in the literature [25–27]. Our approach provides Deep Reinforcement Learning (DRL) training with adaptive levels of task difficulty. In particular, we employ Land-ACL to establish a training curriculum for the agent, enabling it to learn to land on an accelerating and steering UGV under wind interference. In addition to Land-ACL, we design a Landing Vision System (LVS) for UGV localization and pose estimation. An overview of our proposed method is presented in Figure 1.

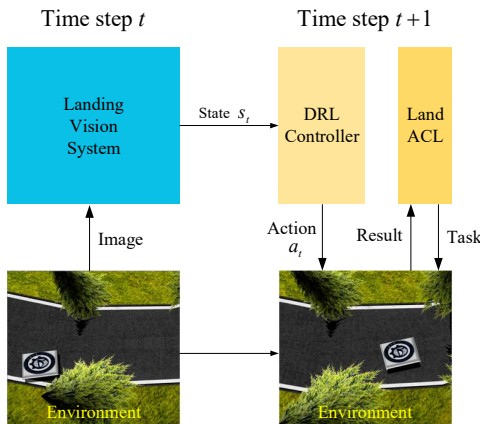

**Figure 1.** An overview of our proposed autonomous landing system.

The main contributions are summarized as follows:

- We propose an automatic curriculum learning framework for solving the UAV-UGV landing problem under different conditions of UGV motions and wind interference.
- We design a task difficulty discriminator to schedule the curriculum according to the levels of difficulty.
- We design a streamlined pipeline that enables rapid visual tracking while providing pose estimation for the DRL controller.

The rest of the paper is organized as follows: Section 2 presents the related preliminaries of deep reinforcement learning and the kinematic model. Section 3 describes the proposed method. Section 4 discusses experimental results. Finally, we conclude the paper in Section 5.

## 2. Preliminaries

### 2.1. Deep Reinforcement Learning

Recent advancement in DRL has led to a variety of algorithms such as Deep Q-learning (DQN) [28], Deep Deterministic Policy Gradient (DDPG) [14], Twin-delayed Deep Deterministic Policy Gradient (TD3) [22] and the Soft Actor Critic (SAC) [29]. We note that SAC demonstrates good learning results by introducing a policy entropy to achieve efficient explorations throughout training. However, the SAC failed to distinguish itself with the performance of TD3 in [30–32]. Therefore, we choose TD3 for our policy training.

TD3 incorporates several key features that enhance its efficiency and stability when compared to DDPG. One significant distinction is the utilization of twin Q-functions, which helps prevent the Q-value from being overestimated. Additionally, TD3 implements a delayed policy update mechanism, where the policy network is updated less frequently than the Q-value networks $Q_{\theta_i}(s,a)_{i=1,2}$, where $\theta_{i(i=1,2)}$ are the parameters of the critic networks. Furthermore, TD3 employs the technique of target policy smoothing, which introduces noises into the learning process to prevent convergence to unfavorable local optima. The Bellman equation is utilized to calculate target value $y(r,s')$. The smaller item of the two outputs $\left(Q_{\theta'_1}, Q_{\theta'_2}\right)$ from the target critic networks is fed into the Bellman equation to avoid overestimation of the Q-value as

$$y(r,s') = r + \gamma \min_{i=1,2} Q_{\theta'_i}(s', \pi_\phi(s') + \varepsilon). \tag{1}$$

We then update the critic networks as follows:

$$\theta_i \leftarrow \text{argmin}_{\theta_i} N^{-1} \sum (y - Q_{\theta_i}(s,a))^2. \tag{2}$$

Meanwhile, the target actor network is not updated consecutively to reduce the overestimation problem. In TD3, however, parameter $\phi$ is updated by the deterministic policy gradient after certain training iterations, e.g., updated two times [22]. The equation for the policy update is described as follows:

$$\nabla_\phi J(\phi) = N^{-1} \sum \nabla_a Q_{\theta_1}(s,a)\Big|_{a=\pi_\phi(s)} \nabla_\phi \pi_\phi(s). \tag{3}$$

Finally, hyper-parameter $\tau (0 < \tau < 1)$ is introduced for the soft update, which alleviates the problem of over-fitting during the training. The target networks are updated as follows:

$$\begin{aligned} \theta'_i &\leftarrow \tau\theta_i + (1-\tau)\theta'_i, \\ \phi' &\leftarrow \tau\phi + (1-\tau)\phi'. \end{aligned} \tag{4}$$

#### 2.1.1. Reward Function Setup

The reward function plays a crucial role in learning an effective and efficient UAV landing. First, it is essential for the UAV to land precisely on top of the UGV, within the specified marker range. Second, the UAV is required to continuously track the UGV to ensure accurate state estimation throughout the landing process, maximizing the visibility

of the marker whenever possible. Lastly, the landing task should be completed as quickly as possible. Consequently, the received reward $r_t$ at each time step is designed as follows:

$$r_t = \begin{cases} 100 & \text{success} \\ r_c - 2 & \text{otherwise} \\ -100 & \text{failed} \end{cases}. \tag{5}$$

Furthermore, we introduce reward item $r_c$ to facilitate the UAV to keep up with the moving target at each time step. The reward associates with the state space, $\mathbb{R}^{3\times3\times1}$, driving the agent to stick to the moving platform. We add $-2$ to the step reward to prevent the agent from learning to hover over the center of the landing zone without landing. $r_c$ is defined as follows:

$$r_c := \sum_{s\in\mathbb{R}} \frac{\text{count}(I)}{\text{count}(I_{\text{total}})}, \tag{6}$$

where $I$ is the number of pixels which exceeds the threshold value, and $I_{\text{total}}$ represents the total number of pixels in $\mathbb{R}^{3\times3\times1}$. The value of $r_c$ has a range of $[0,1]$ to encourage the agent to approach the center point of the UGV at each time step.

### 2.1.2. TD3-Based Landing Controller

The control process starts with the UAV capturing an image of the environment using an on-board camera. The camera's orientation is adjusted to encompass the target landing area, and the image is captured with appropriate exposure and focus settings. The state vector consists of features generated by the Landing Vision System (LVS), which we introduce later in Section 3.1. At time step $t$, state vector $s_t$ is defined as $s_t = \mathbb{R}^{9\times1\times1}$. The image is then fed into the TD3 network, which extracts relevant information and generates an action denoted as $a_t$. Action space $A$ is defined as a three-dimensional continuous space. Specifically, action $a$ is denoted as $a = (v_x, v_y, v_z) \in A$, where $v_x$, $v_y$, and $v_z$ represent the reference linear velocities along the $x$, $y$, and $z$ axes in the world coordinate system, respectively. The calculation of $a_t$ takes into account the UAV's current state, including its position, velocity, and the processed image data. The TD3 network effectively combines reinforcement learning with deep neural networks to optimize the landing process. By learning from previous landing trials, it dynamically adapts its strategies in real time, ensuring smooth landing.

### 2.2. Coordinate System and the UAV Kinematics

The coordinate system includes the Rack and Camera Coordinate System along with Pixel, World and the Target Coordinate Systems, as shown in Figure 2.

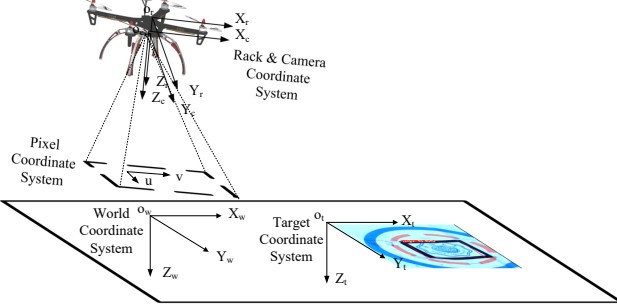

**Figure 2.** The definition of the coordinate system.

We use a quadrotor with an X-configuration rack. Four mini-motors are installed perpendicular to the rack to provide up-lifting forces $T_{i(i=1,2,3,4)}$ through rotation in the $O_rZ_r$ negative direction. The rack coordinate system and moments along with forces are also shown in Figure 3.

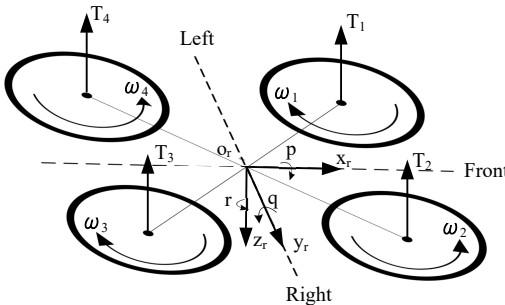

**Figure 3.** Coordinate systems and forces acting on the UAV.

We assume that the UAV is subject to gravity in the $O_w Z_w$ only and that its frame does not deform during flight (i.e., it is rigid). The kinematics of the UAV are shown as follows:

$$m\ddot{z} = -mg + \sum_{i=1}^{4} T_i, \tag{7}$$

where $m$ is the mass of the UAV, $\ddot{z}$ represents the acceleration in the z-coordinate, $g$ is the gravitational acceleration, and $T_i$ represents the thrust generated by the $i$th propeller. This equation accounts for the gravitational force and the thrust from the propellers in maintaining the UAV's altitude. The horizontal motion in the *x*-axis and *y*-axis is described by

$$m \begin{bmatrix} \ddot{x} \\ \ddot{y} \end{bmatrix} = T \cdot \begin{bmatrix} \sin(\phi) \cdot \cos(\theta) \\ \sin(\phi) \cdot \sin(\theta) \end{bmatrix}, \tag{8}$$

where $\ddot{x}$ and $\ddot{y}$ represent the acceleration in the *x*-axis and *y*-axis. $T$ is the total thrust generated by all propellers, $\phi$ is the pitch angle, and $\theta$ is the roll angle.

We can conclude the kinematic model of the UAV as follows:

$$\begin{cases} \begin{bmatrix} \dot{X}^w \\ \dot{Y}^w \\ \dot{Z}^w \end{bmatrix} = \begin{bmatrix} v_x^w \\ v_y^w \\ v_z^w \end{bmatrix} \\ \begin{bmatrix} \dot{\phi} \\ \dot{\theta} \\ \dot{\psi} \end{bmatrix} = \begin{bmatrix} 1 & \tan\theta\sin\phi & \tan\theta\cos\phi \\ 0 & \cos\phi & -\sin\phi \\ 0 & \sin\phi/\cos\theta & \cos\phi/\cos\theta \end{bmatrix} \begin{bmatrix} p \\ q \\ r \end{bmatrix} \end{cases}, \tag{9}$$

where $(X^w, Y^w, Z^w)^{\mathrm{T}}$ is the position of the UAV in the World coordinate system.

The UGV moves in the $O_w Y_w X_w$ plane, and we denote its position in the World coordinate system as $(X^t, Y^t, Z^t)^{\mathrm{T}}$. Therefore, we can calculate the relative position between the UAV and UGV as follows:

$$\begin{bmatrix} X_D^w \\ Y_D^w \\ Z_D^w \end{bmatrix} = \begin{bmatrix} X^w \\ X^w \\ Z^w \end{bmatrix} - \begin{bmatrix} X_t^w \\ X_t^w \\ Z_t^w \end{bmatrix}. \tag{10}$$

## 3. Method

### 3.1. Landing Vision System

The Landing Vision System (LVS) is designed to enable the UAV to recognize and locate the target UGV, as depicted in Figure 4. First, the UAV captures an RGB image of $81 \times 81$ pixels using a downward-facing camera attached to the bottom of the UAV. These visual inputs are then fed into the Landing Vision Module (LVM) for target recognition and visual tracking. Within the LVM, we initially employ the Oriented FAST and Rotated BRIEF (ORB) algorithms [33] and the Brute Force Matcher to detect the marker by comparing it to a provided template. A bounding box is drawn based on the matching result. We then highlight the area within the bounding box and darken the surrounding area. If the UGV

moves out of the Field Of View (FOV) of the UAV, we aim to restore the state of the previous detected time step. If ORB fails to detect the UGV at the current time step, we replace the current state with the latest state that includes the UGV's location. Then, we employ the Depthwise Separable Convolution (DSC) method [34] to localize the UGV within a single image frame. Then, we normalize the output image and utilize the Ghosting method to provide the agent with the UGV's motion trends. This approach in the LVM assists the agent in adapting to real-world target detection without overwhelming the DRL agent with excessive data. Finally, the RGB image is converted into a grayscale image and then reshaped into a $9 \times 1$ array, constituting the state vector at time step $t$.

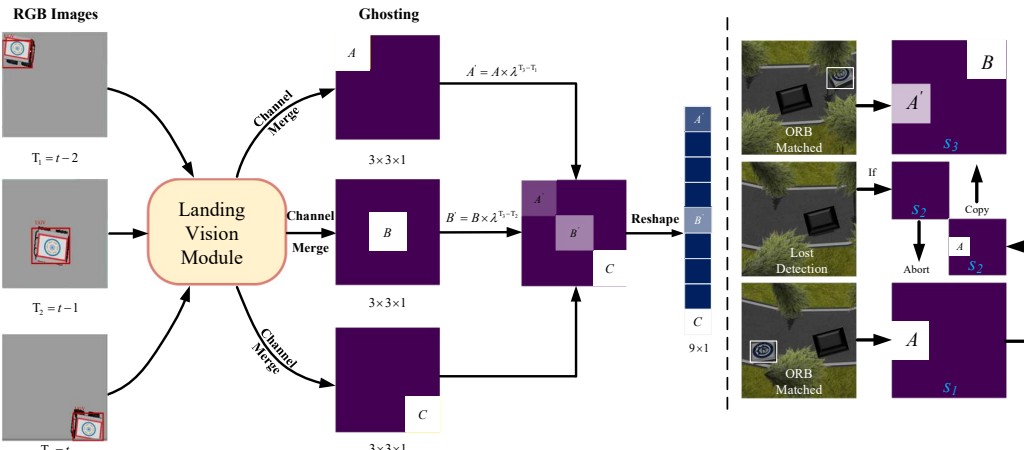

**Figure 4.** Pipeline of the Landing Vision System. A, B and C are the corresponding pixel values of the ground target in the images, which are changed into A', B' and C' using the proposed Ghosting method.

We note that the Ghosting method extracts the UGV's motion trend by analyzing a sequence of consecutive frames. The UGV's location is extracted by the ORB algorithm and a Brute Force Matcher. We use the previously prepared feature points extracted from the template of the marker that guides the landing process. After the relative location of the UGV is calculated, we highlight the area that contains the UGV and use DSC to divide the approximate position of the UGV into the corresponding area of the final $3 \times 3$ image. We record the pixel value within the collected images at the time step $n$ as $p_n$ and adjust the pixel value $p_n$ within the collected images as follows:

$$p'_n = p_n \cdot \lambda^{t-n}, \tag{11}$$

where a discounting variable $\lambda$ ($\lambda \in (0, 1)$) is introduced to concatenate continuous frames into a single frame. As the frame's temporal proximity to the current time step $t$ increases, so does its capacity to preserve higher pixel values. Consequently, the composite result of aggregating these processed images elegantly encapsulates the evolving motion trends of the UGV. This not only provides the UAV with a comprehensive understanding of the UGV's motion state, but also leverages this knowledge to guide subsequent decisions and actions.

### 3.2. Automatic Curriculum Learning of the Landing Policy

#### 3.2.1. Task Setup

We divide the landing task into three steps, each featuring varying levels of difficulty, as shown in Table 1.

**Table 1.** Task Item of Land-ACL.

| Task | Parameters | Description |
|------|------------|-------------|
| *Basic* | $v_{UGV}$ | Maximum linear velocity of UGV |
| *Default* | $a_{UGV}, \omega_{UGV}$ + *Basic* | Maximum angular velocity and acceleration of UGV |
| *Frontier* | $v_{wind}$ + *Default* | Maximum wind speed |

- *Basic*: We introduce the *Basic* task to provide the agent with primary signals for learning low-level principles of landing on the UGV. Specifically, the *Basic* task contains a single variable: the UGV's linear velocity $v_{UGV}$. By introducing the *Basic* task, our aim is to facilitate the agent to learn basic landing strategies, as shown in Figure 5. Moreover, the action policy learned in *Basic* is intended to prepare the agent for the transition from *Basic* to the subsequent *Default* task, addressing the gap that may arise during task switching.

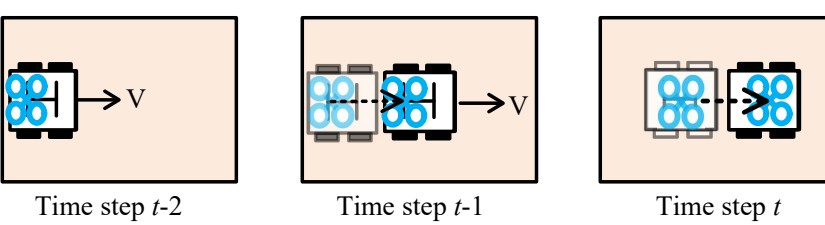

Time step *t*-2      Time step *t*-1      Time step *t*

**Figure 5.** Illustration of a *Default* task.

- *Default*: The *Default* task is designed to facilitate learning to land in complex situations. The *Default* task incorporates two additional variables based on the *Basic* task: angular velocity $\omega_{UGV}$ and speed acceleration $a_{UGV}$, as illustrated in Figure 6. We change the direction of $\omega_{UGV}$ at regular intervals. We intend to enhance the agent's ability for quick response when dealing with unexpected environmental changes. In addition, the robustness of the landing process can also be improved through adding acceleration and steering.

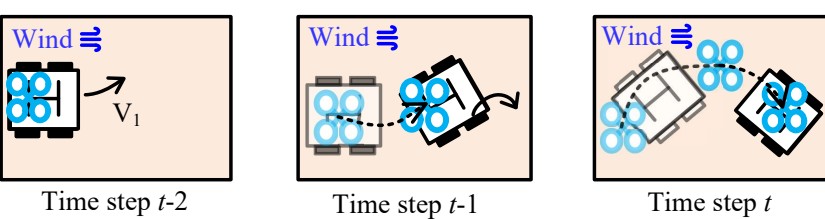

Time step *t*-2      Time step *t*-1      Time step *t*

**Figure 6.** Illustration of a *Frontier* task.

- *Frontier*: In order to further enhance the adaptability to complex environments for the policy, we introduce the *Frontier* task to facilitate learning to land with wind interference, which is simulated using continuous Gaussian noise added to the UAV's maneuver commands. This Gaussian noise $v_{wind}$ is limited to a magnitude of 0.3 m/s and lasts for a maximum of 2 s. Furthermore, we use the augmentation of $v_{wind}$ as a simulated-to-real strategy to prepare the policy for handling environmental disruptions during real flights.

### 3.2.2. Land-ACL

We propose the Land-ACL method to tackle the challenging problem of landing a UAV on a moving UGV. Traditional manual curriculum design is inefficient for addressing this high-dimensional control problem that demands a complex skill set for enabling precise motion control in dynamic environments [21]. To solve this problem, our Land-ACL method automatically generates curriculums of increasing difficulty based on the agent's learning progress, as illustrated in Figure 7.

At the beginning of a training episode, we randomly generate multiple tasks *U* from *Basic*, *Default* and *Frontier*. Next, we employ a neural network discriminator to evaluate whether the task difficulty is appropriate for the agent to manage. Using the received task parameter values, the Difficulty Discriminator (DD) can produce the agent's success probability *P* for completing the current task. Our DD training employs the end-to-end training method, entailing the acquisition of knowledge from scratch. By intentionally configuring a replay buffer with limited capacity, the curriculum learning enables Land-ACL to focus on guiding recent task learning. This approach enables efficient learning by progressively increasing the difficulty of the learning tasks, ensuring that the agent steadily acquires the essential skills for successful landings. The structure of our Difficulty Discriminator can be seen in Table 2.

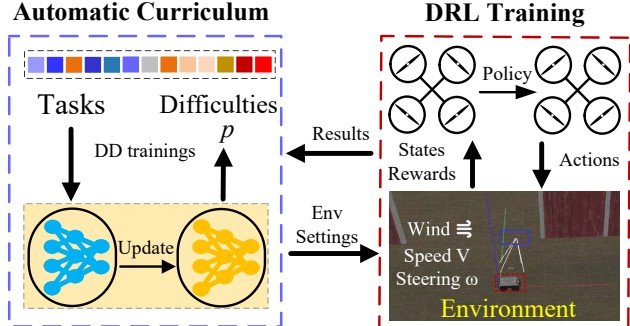

**Figure 7.** Pipeline of the Landing-Automatic Curriculum Learning method.

**Table 2.** Difficulty Discriminator for Land-ACL

| Input | Network Dimensions | Output | Operator |
|---|---|---|---|
| $V_{UGV}$, $\omega_{UGV}$ | $3 \times 15$ | FC1 | Leaky Relu |
| FC1 | $15 \times 8$ | FC2 | Leaky Relu |
| FC2 | $8 \times 1$ | *P* | Tanh |

We then filter the output task based on success probability, ensuring a consistent challenge for the agent with tasks of balanced complexity. If *P* falls within a specific threshold range ($\eta$, $\xi$), we determine that this task has the appropriate difficulty for the current stage of learning. The environment is then reset based on the output task variables. Finally, we configure the new environment according to the output task *U*, as illustrated in Algorithm 1. The difficulty is predicted by DD, even if it is not properly trained. We test the current capability of the agent every 10 episodes using the critical difficulty of the task type. Once the agent passes the test, the task type switches to the next one.

---

**Algorithm 1** Land-ACL

---
**Input:** Order;
**Output:** Task $U$;
 1: **for** $t = 1$ to $T$ **do**:
 2:     **if** $t \bmod 10 = 0$ **then**:
 3:         Conduct landing capability Tests;
 4:     **end if**;
 5:     $TaskType \leftarrow GetTaskType()$;                              $\triangleright$ The task type changes if the agent passes the capability test
 6:     **while** True **do**:
 7:         **if** $TaskType = Basic$ **then**:
 8:             Obtain $U$ from $GetTaskBasic()$;
 9:         **end if**;
10:         **if** $TaskType = Default$ **then**:
11:             Obtain $U$ from $GetTaskDefault()$;
12:         **end if**;
13:         **if** $TaskType = Frontier$ **then**:
14:             Obtain $U$ from $GetTaskFrontier()$;
15:         **end if**;
16:         Randomize parameters in $U$;
17:         Clip $U$ according to the boundaries;
18:         $p \leftarrow DD(U)$;                                        $\triangleright$ Obtain the success probability of the output task.
19:         **if** $\eta < p < \xi$ **then**: **return** $U$.
20:         **end if**;
21:     **end while**
22: **end for**.

---

## 4. Experiments and Results

The experiment was conducted in the Gazebo simulation environment. MAVLink was used to connect the PX4 flight controller with the simulator. Additionally, we modified the Iris-type UAV from the firmware package by installing a bottom-mounted camera, oriented perpendicular to the frame. We note that the downward-facing camera was rigidly attached to the UAV's rack to reduce possible vibrations generated through the motor rotations. The Field Of View (FOV) for the equipped wide-angle First Point of View (FPV) camera has a limited angle of 120 degrees. We set the simulated experiment in clear lighting conditions for UAV visual recognition.

For the design of the ground target, we imported the Husky UGV model and made modifications to its top plate to accommodate the detection marker, as depicted in Figure 8. The marker was based on the logo of Hohai University. The speed and steering angle limits of the Husky UGV were customized to meet the training and testing needs.

Our experiment was divided into two phases: training and testing. We employed the Adam optimizer [35] in the neural network, with a learning rate of $\alpha = 10^{-4}$ for the actor network and $\alpha = 10^{-3}$ for the critic network. We used a discount factor $\gamma = 0.99$ to compute the expected Q-value. To facilitate soft updates, we set $\tau = 0.005$. Rectified linear units (ReLUs) were used for all hidden layers in the neural networks. The actor's final output layer was a tanh layer to constrain the actions. The agent was trained using a mini-batch size $N$ of 16, with 10 iterations conducted whenever the UAV landed. Both replay buffers, $B$ and $B_v$, had a size of 32,768.

### 4.1. Simulated Trainings

During training, the UAV first ascended to a height of 3.5 m, followed by the initiation of the unmanned ground vehicle (UGV). During the training, the speed of the UGV was set to vary from 0.2 m/s to 1 m/s. Furthermore, we introduced a sudden acceleration of the UGV with a value of $a_{ugv} = 0.2$ m/s$^2$. We conducted 10 trials for the policy training, with each trial comprising 400 episodes.

We updated the two critic networks and one actor network of TD3 five times per iteration. For fair comparisons, we conducted experiments on SAC, TD3 and Land-ACL+TD3 algorithms under similar conditions. We calculated the average accumulated reward obtained during the training trials for all three algorithms, as shown in Figure 9a. The loss curve for our DD training is shown in Figure 9b. It illustrates that our proposed algorithm achieved the fastest convergence, reaching stability after around 150 episodes. Additionally,

the cumulative reward of our algorithm steadily increased until it reached a plateau after approximately 350 episodes of training. In other words, the proposed algorithm outperformed the classical TD3 approach in terms of the final cumulative reward. This indicates the effectiveness of Land-ACL in providing superior training resources for identifying the optimal landing approach rather than inefficient exploration. We note that the SAC algorithm showed worse performance than TD3, which was also reported in [31]. We attribute this common phenomenon to the SAC algorithm's excessive focus on environmental exploration tasks, resulting in poor performance in gaining more accumulated reward in the early stages of training. This phenomenon might be improved through extensive training.

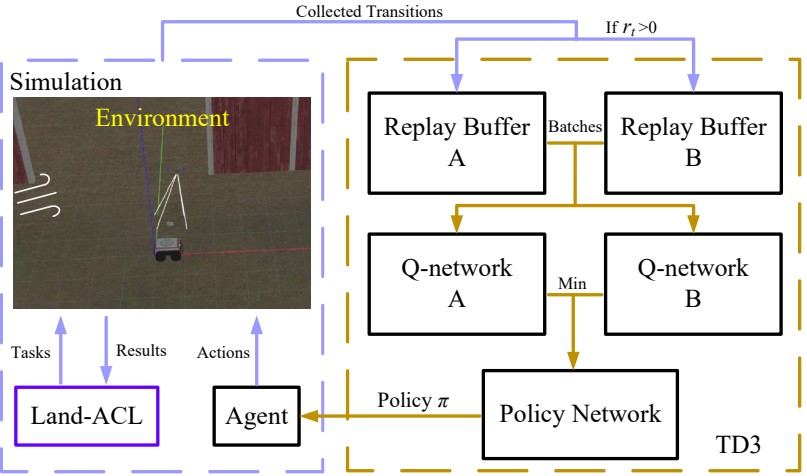

**Figure 8.** Pipeline of our TD3 landing controller.

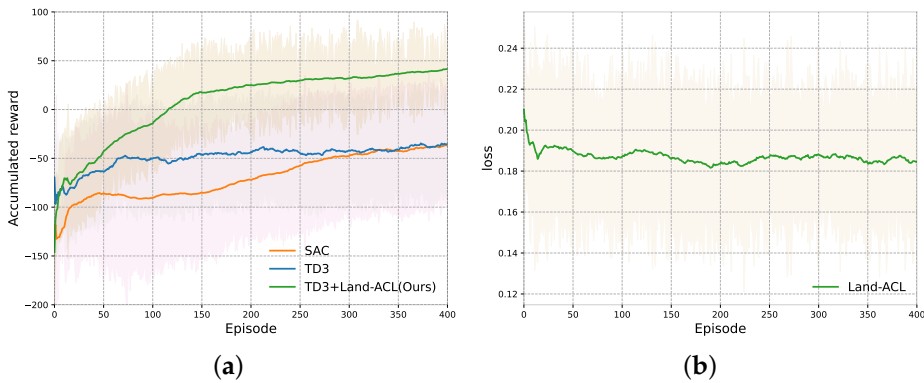

**Figure 9.** Comparison of averaged accumulated reward and the averaged loss curve of the TD3+Land-ACL. (**a**) Learning curve for the averaged accumulated reward. (**b**) Averaged loss curve of DD.

The reward of the Land-ACL+TD3 exhibited a significant increase around the 50th episode and consistently maintained a higher level compared to the other algorithm. In contrast, TD3 did not achieve satisfactory performance within the 400 episodes of training. One contributing factor to TD3's lower reward was the inclusion of a negative reward as specified in Equation (6). Specifically, if the UAV lost track of the UGV within its field of view, a negative reward was incurred. Furthermore, once the UAV lost track, reestablishing the detection of the UGV was challenging. Consequently, the UAV continued to receive negative rewards until the completion of a failed trial. Although it is possible that TD3 could have achieved better performance with additional training episodes, allocating computational resources and time for this purpose would be unnecessary.

*4.2. Simulated Testings*

To assess the effectiveness of our policies under various scenarios, we conducted experiments in two testing scenarios A and B. The results are shown in Table 3. Since

the policies trained with TD3 were suboptimal, indicating their inability to fulfill the *Default* task, we implemented only the tracking policy trained under TD3+Land-ACL. We conducted testings under two scenarios, each tested for 100 trials. Following the approach of Rodriguez-Ramos et. al. [36], we considered a landing successful if the UAV landed 0.1 m above the moving UGV within a distance limit of 0.8 m.

**Table 3.** Statistics for the Simulated Tests.

| Scenario | Average Distance Error (m) | | Total | Average Velocity Error (m/s) | | Total |
|----------|-------------|-------------|-------|-------------|-------------|-------|
|          | *x*-Axis    | *y*-Axis    |       | *x*-Axis    | *y*-Axis    |       |
| Scenario A | $0.45 \pm 0.05$ | $0.34 \pm 0.04$ | 0.56 | $1.12 \pm 0.04$ | $0.52 \pm 0.03$ | 1.23 |
| Scenario B | $0.47 \pm 0.07$ | $0.40 \pm 0.05$ | 0.62 | $1.15 \pm 0.04$ | $0.57 \pm 0.01$ | 1.28 |

4.2.1. Test Scenario A: UGV Moving along a Straight Trajectory

We set the UGV to follow a straight trajectory at a constant speed of 0.6 m/s until reaching the destination, as depicted in Figure 10a. In Scenario A, our primary focus was on testing the UAV's ability to complete the *Basic* task. Consequently, we included only a uniform linear velocity in Test Scenario. The UAV successfully tracked the trajectory of the target UGV while planning smooth landing curves in all of the testing trials using Land-ACL-trained policy, as depicted in Figure 10b. Furthermore, we achieved a 100% success rate in all 100 tests, demonstrating satisfying stability with our proposed method. This demonstrates TD3+Land-ACL's advantages in handling *Basic* and *Default* tasks, resulting in improved motion stability and safety throughout the tracking and landing process.

We also collected the averaged distance and velocity errors from the testing trials, as shown in Figure 10c,d. We can conclude that both distance errors and velocity errors achieved a satisfying outcome for the UAV autonomous landing task. To further challenge the boundaries of our proposed method, we conducted additional testing under Scenario B to fulfill the *Frontier* task.

4.2.2. Test Scenario B: UGV Moving along a Curved Trajectory

To further validate the capability of our method in completing the *Frontier* task, we configured the UGV to follow a curved trajectory with sudden acceleration and wind interference. We simulated high-wind conditions for landing by introducing continuous Gaussian noise into the UAV's action commands. The supplementary action noises were limited to a magnitude of 0.2 m/s and lasted no longer than 2 s. Random sudden velocity changes, ranging from $-0.2$ m/s$^2$ to 0.2 m/s$^2$, were applied to the UGV's motion every 5 s during the training trials. The setup for Test Scenario B is depicted in Figure 11a.

Similarly, the 3D trajectory of the UAV and the UGV is shown in Figure 11b. Compared to the landing trajectories in Figure 10b, the output curves exhibit greater instability. We interpret this phenomenon as the UAV attempting to predict the possible movement pattern of the accelerating and steering UGV.

The averaged distance and velocity errors from the testing trials are collected during each test episode, as depicted in Figure 11c,d. The averaged distance error is relatively small and stable during most of the testing steps, gradually converging to a landing on top of the UGV by the end. However, the averaged velocity error fluctuated significantly during the landing process. This phenomenon can be attributed to the UAV attempting to keep up with the UGV. The positive–negative variation in Figure 11d represents the process of the UAV quickly adjusting to the changing UGV speed. Despite the fluctuations, our method achieved a 91% success rate in the testing trials under Scenario B, demonstrating that TD3+Land-ACL is capable of completing the *Frontier* task, therefore further demonstrating the effectiveness of our proposed Land-ACL in guiding the UAV's landing on a speed-changing UGV under wind interference.

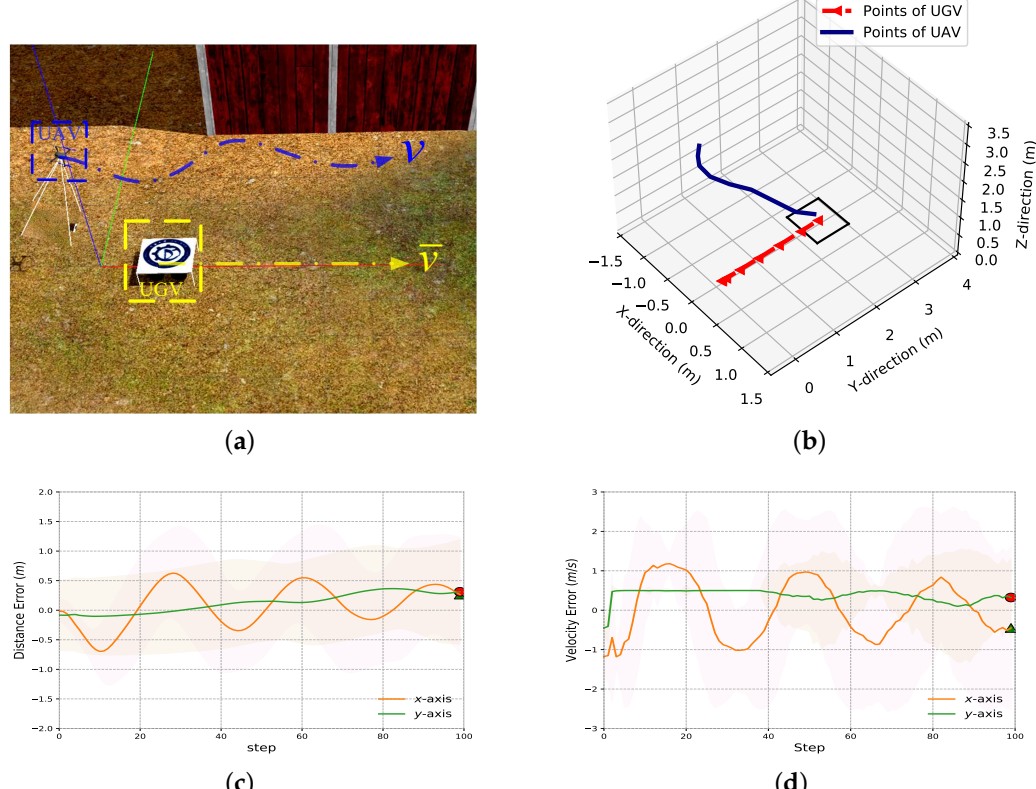

**Figure 10.** Illustration of the averaged trajectory, distance and velocity error for our TD3+Land-ACL in Test Scenario A. (**a**) Training diagram of Scenario A in the Gazebo simulator. (**b**) Averaged trajectories of tracking a moving UGV. (**c**) Averaged distance error. (**d**) Averaged velocity error.

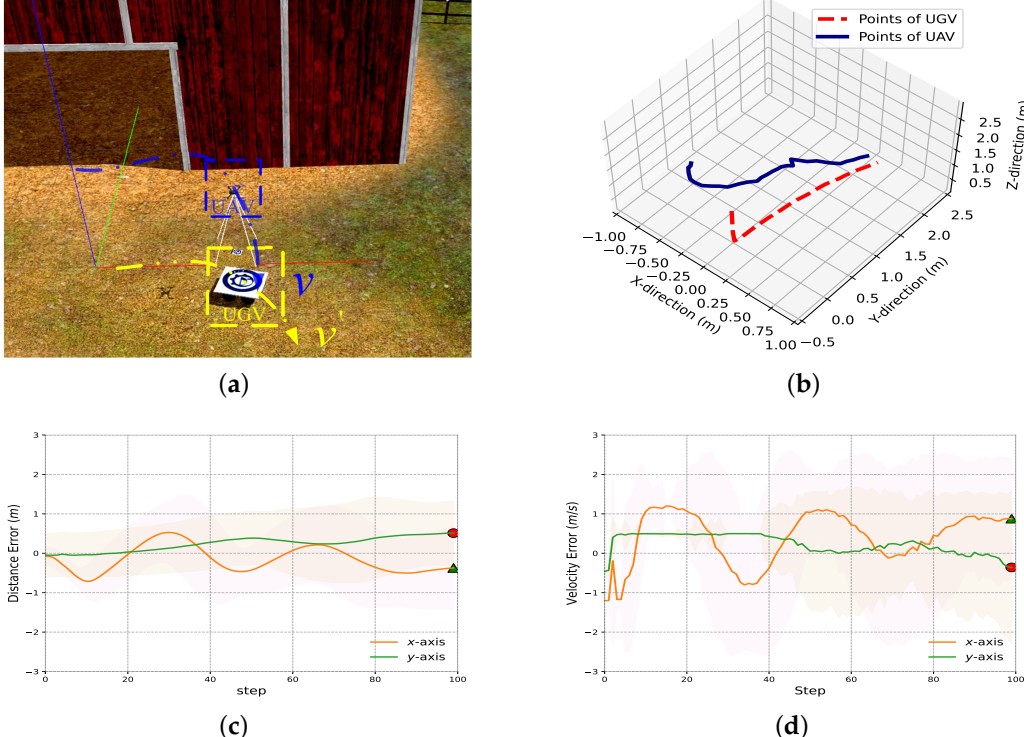

**Figure 11.** Illustration of the averaged trajectory, distance and velocity error for our TD3+Land-ACL in Test Scenario B. (**a**) Training diagram of Scenario B in the Gazebo simulator. (**b**) Averaged trajectories of tracking a moving UGV. (**c**) Averaged distance error. (**d**) Averaged velocity error.

### 4.2.3. Landing Precision

We collected the successful landing points of 100 trials for each of the three algorithms under Test Scenario B. The tests continued until the testing controller reached 100 successful landing trials. A safe and precise landing should be close to the center of the UGV, i.e., the center of the landing marker. The statistics of the landing point distribution are shown in Table 4. Both the SAC and the TD3 controller exhibit unsatisfying performance with a successful rate of 24% and 36%, respectively. Our method reached a 91% success rate in these testing trials, indicating the trained policy reached satisfying stability. Furthermore, the TD3+Land-ACL demonstrates a more precise and concentrated landing points, with a average distance of 0.44 m and a distribution of 0.2 m. This indicates that our method has good stability and landing accuracy.

**Table 4.** Statistics of the landing Points

| Algorithm | Average Distance (m) | $\sigma$ (m) | Successful Landing Rate (%) |
|---|---|---|---|
| SAC | 0.86 | 0.33 | 24 |
| TD3 | 0.78 | 0.31 | 36 |
| TD3+Land-ACL | 0.44 | 0.20 | 91 |

In Figure 12, we illustrate the landing points generated by a specific algorithm if its landings were successful. The landing points of TD3+Land-ACL are distributed in the lower right part of the landing area. This indicates that the UAV manages to discover better landing policies by predicting the movement of the UGV and catching up with it due to the use of the Ghosting method as part of the DRL state representation. In contrast, the landing points of SAC and TD3 are distributed around the left part of the landing platform, indicating that they do not learn an optimal policy adapting to the movement of the UGV.

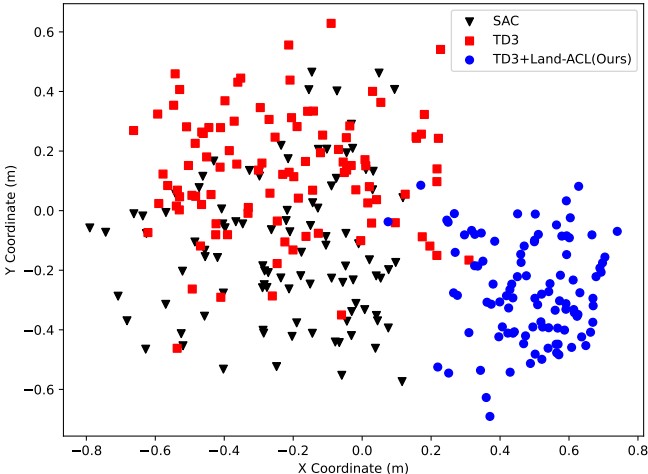

**Figure 12.** Distribution of the landing points on the landing marker.

## 5. Conclusions

We proposed a novel approach that combines the Automatic Curriculum Learning (ACL) method with the Twin Delayed Deterministic Policy Gradient (TD3) algorithm. By incorporating ACL, we provided a policy adaptive to environmental changes for UAV landing control. Additionally, we proposed the Landing Vision System (LVS) to provide constant detection and tracking of the target UGV using the Oriented FAST and the Rotated BRIEF (ORB) algorithms. We proposed the Ghosting method to consolidate the motion trajectories of the moving UGV from multiple images into a single image. Through comprehensive testing in simulations, we validated the effectiveness of our algorithm in adapting to dynamic landing scenarios. The UAV demonstrated a strong adaptability to

interference, and the UAV learned to predict the movement of the UGV for reliable landing. The simulation results showed a satisfactory result of a 91% success rate and a distance error of 0.44 m.

In future work, we will focus on further improving the accuracy of our landing system and carry out real-world experiments.

**Author Contributions:** Conceptualization, C.W. (Chang Wang); methodology, C.W. (Chang Wang); software, J.W.; validation, C.W. (Chang Wang); formal analysis, C.W. (Chang Wang); investigation, C.W. (Chang Wang); resources, C.W. (Changyun Wei); data curation, J.W.; writing original draft preparation, J.W.; writing—review and editing, C.W. (Chang Wang) and Y.Z.; visualization, J.W.; supervision, C.W. (Chang Wang); project administration, D.Y.; funding acquisition, J.L. All authors have read and agreed to the published version of the manuscript.

**Funding:** This work was supported in part by the Science and Technology Innovation 2030-Key Project of "New Generation Artificial Intelligence" under Grant 2020AAA0108200 and in part by the National Natural Science Foundation of China under Grant 61906203 and 62006121.

**Data Availability Statement:** Not applicable.

**Conflicts of Interest:** The authors declare no conflict of interest.

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
