# Peer review of "Vision-Based Deep Reinforcement Learning of UAV-UGV Collaborative Landing Policy Using Automatic Curriculum"

_drones, doi:10.3390/drones7110676_

Round 1

Reviewer 1 Report

Comments and Suggestions for Authors

This article uses a new algorithm to address the collaborative autonomous landing of Unmanned Aerial Vehicles (UAVs) on Unmanned Ground Vehicles (UGVs). The obtained results are interesting, and only minor revisions are needed:

1.     More tasks can be added to enhance adaptability to complex environments.

2.     The paper contains some grammatical errors, and there are issues with the use of English tenses.

3.     To ensure the relevance and currency of the work, it is necessary to include more recent papers within the past three years.

Comments on the Quality of English Language

Minor editing of English language required.

Author Response

Dear Reviewer,

Thank you very much for giving us a chance to revise our paper. We would like to thank you for the thoughtful and thorough review. We have carefully considered your comments and revised the manuscript, and our responses are attached. Hopefully, we have answered all the concerns. The modified or added contents are highlighted in the revised manuscript. We would appreciate your acceptance of the paper for publication in Drones. Once again, thanks for your time and efforts. We are looking forward to hearing from you.

Yours Sincerely,

Jiaqing Wang

Reviewer 2 Report

Comments and Suggestions for Authors

1. Although TD3 algorithm is a commonly used algorithm, it was proposed in 2018 and has been around for 5 years now. Therefore, TD3 should not be used as a benchmark algorithm for comparison.

2. The article uses curriculum learning, but currently there are many studies related to reinforcement learning that have introduced curriculum learning to accelerate the training process. The author should address the differences between the algorithms proposed in the article and these algorithms, such as:

[1] Z. Ren, D. Dong, H. Li and C. Chen, "Self Paced Prioritized Curriculum Learning With Coverage Penalty in Deep Retirement Learning," in IEEE Transactions on Neural Networks and Learning Systems, vol. 29, no. 6, pp. 2216-2226, June 2018, doi: 10.1109/TNNLS.2018.2790981

[2] Z. Hu, X. Gao, K. Wan, Q. Wang and Y. Zhao, "Asynchronous Curriculum Experience Replace: A Deep Reinforcement Learning Approach for UAV Autonomous Motion Control in Unknown Dynamic Environments," in IEEE Transactions on Vehicle Technology, doi: 10.1109/TVT.2023.3285595

3. The TD3 algorithm is introduced in section 3.3 for the first time, but it is shown in Figure 1 in the introduction section. This will bring confusion to readers.

4. The introduction to the simulation environment used should be more specific to facilitate readers' reproduction of the paper.

5. Personally, I suggest replacing Figure 12 with a scatter plot.

6. How long does DD training take to ensure accuracy?

7. The grammar errors in the paper need to be corrected.

Comments on the Quality of English Language

The grammar errors in the paper need to be corrected

Author Response

(The authors gave the same response as above.)

Reviewer 3 Report

Comments and Suggestions for Authors

The paper presents a DRL method for autonomous landing in moving platforms. The scope of the paper is really interesting for the Drones journal, and the results are interesting, but should be improved to a great extent before publication.

In this case, the authors propose an Automatic Curriculum learning procedure Land-ACL that would help the training process and obtain better results. The idea is to start learning in straightforward examples and then generate more difficult ones in latter stages of the learning process.

A strong point of your approach is that you consider real marker detection. However, you don't specify the exact method and the Ghosting methodology is not defined in a clear way. Please state which marker are you using and the properties of your marker detection method. Have you considered an all-learned method in which the marker estimator that estimates the UGV pose is also learned?

In the Preliminaries section, it seems that you only consider dynamics on Z coordinate. However, the equations might suggest that z'w  = 0 --> the UAV flights at a constant altitude. Similarly, eq. (1) might suggest that vx and vy are zero. Please, correct it, by separating the z part of the x and y part (one has a dynamic model, the other a kinematic one).

When estimating the UGV pose, you use the Depthwise Separable Convolution method (DSC), which is neither explained or cited. Also, you claim to use a high-resolution camera of 81 pixels? How on Earth is that High-res?

Also, the description of the Ghosting method is also vague. You claim to put a discounting variable for forgetting past images. However, do you consider the motion of the UAV to translate/rotate/scale past images?

Your method uses a Twin Delayed Policy Gradient? (TD3) or Deep Deterministic? You name it differently in the Introduction and in the Conclusion sections. Moreover, as your approach strongly depends on this method, it would be great if you introduce it in the DRL method, and then particularize the state vector and everything once defined. In this way, it would be more clear.

Regarding the reward function. To me is not clear the range of values of rc (it seems to belong to [0,1]), and why do you substract 2 to it in the reward function?

Regarding the Difficulty Discriminator (DD), it seems intersting but it is not clear to me how the difficulty is predicted at a first stage. Besides, how do you decide the task type when generate. Additionally, in Algorithm 1 it is not clear the purpose of the outer for loop (as defined, the function should only return one task).

The experiment part is very nice, with a great set of experiments performed and where the details of the parameters of the learning process on both DD and Land-ACL available. However, it would be great if you include a public repository in which the interested reader could reproduce your experiments. In addition, I think that the structure of the DD network should be specified when it is being described in the previous section.

Unfortunately, the section is plagued with typos on unit types (accel. is m/s^2). Distance limit in square meters?? YOur method performs very well in straight line, however I have a minor issue with Scenario B: why do you do punctual accelerations each 5 seconds, rather than constant accel. which should give smoother trajectories. In particular, the UGV should cooperate, not perform random movements.

Comments on the Quality of English Language

The paper presents a significant number of typos and not revised sections. Therefore, it is recommended to be proof-readed by an English native person before next submission.

Some typos:

- citeref16 (page 2)

- line 111: contiunes (consecutive?)

- line 131: "a" continuous (missing a)

- The unit measures of section 4 should be revised

-line 292: scenairo

- line 312: last sentence is not gramatically correct

Author Response

(The authors gave the same response as above.)

Reviewer 4 Report

Comments and Suggestions for Authors

Dear Authors,

The proposed paper is interesting and brings some information on the new method of controlling the UAV landing on the moving noncooperative. That method seems to be a perspective one. It would be good to train the network using real-life data. Of course, it is not a matter of your research and the paper itself but future research.

The way of presentation of the referred literature, applied methods and achieved results is very comprehensive and easy to understand by the readers. I would suggest to extend only one paragraph, i.e. the Conclusions. You have summarised the work done during the performed research but the achieved results. It is necessary to add the summary of the achieved results, which were presented in the paper.

In my opinion, the paper, after adding that extension can not be published in its present form in that magazine.

Author Response

(The authors gave the same response as above.)
